# The Inducible Intein-Mediated Self-Cleaving Tag (IIST) System: A Novel Purification and Amidation System for Peptides and Proteins

**DOI:** 10.3390/molecules26195948

**Published:** 2021-09-30

**Authors:** A. Sesilja Aranko, Hideo Iwaï

**Affiliations:** Institute of Biotechnology, HiLiFE, University of Helsinki, P.O. Box 56, 00014 Helsinki, Finland; sesilja.aranko@aalto.fi

**Keywords:** halophilic intein, NMR, protein splicing, conditional protein splicing, C-terminal amidation, isotopic labeling, peptide purification, purification tag, self-cleaving tag

## Abstract

An efficient self-cleavable purification tag could be a powerful tool for purifying recombinant proteins and peptides without additional proteolytic processes using specific proteases. Thus, the intein-mediated self-cleavage tag was developed and has been commercially available as the IMPACT™ system. However, uncontrolled cleavages of the purification tag by the inteins in the IMPACT™ system have been reported, thereby reducing final yields. Therefore, controlling the protein-splicing activity of inteins has become critical. Here we utilized conditional protein splicing by salt conditions. We developed the inducible intein-mediated self-cleaving tag (IIST) system based on salt-inducible protein splicing of the MCM2 intein from the extremely halophilic archaeon, *Halorhabdus utahensis* and applied it to small peptides. Moreover, we described a method for the amidation using the same IIST system and demonstrated ^15^N-labeling of the C-terminal amide group of a single domain antibody (V_HH_).

## 1. Introduction

Many bioactive peptides, such as peptide hormones, antibacterial peptides, and growth factors, are amidated at the C-terminus. The C-terminal amidation is often required for their total activity or prolonged bioavailability [1,2]. Therefore, a method for the amidation of bioactive peptides can be a valuable tool for pharmaceutical industries. In particular, recombinant proteins and peptides from prokaryotes require additional enzymatic modification steps for the C-terminal amidation [1,2]. Intein-mediated self-cleavages have been applied for developing purification tags because self-cleavable tags could eliminate proteolytic removal steps of purification tags [3,4,5,6,7]. The intein-mediated purification with an affinity chitin-binding tag (IMPACT™) system has been commercially available from New England Biolabs (Ipswich, USA). The IMPACT system exploits self- and auto-catalytic inteins with Ala mutation at the last residue so that the protein splicing is halted after the first N-S acyl shift (Figure 1a, (II)) [3,4,5]. The thioester intermediate can be released by a thiol agent or hydrolysis [3,4,5]. The intein-mediated self-cleavage tag approach using the IMPACT™ system was also applied to the carboxyl amidation of peptides [8]. Ammonium ions function as nucleophiles to modify the α-thioester group of the amides [8] (Figure 1a). Therefore, C-terminal amidation using the IMPACT™ system could be a convenient method for recombinant peptides [8]. However, the intein-mediated approach using the IMPACT™ system or other inteins could be limited by the C-terminal amino-acid type of the target protein/peptide because premature cleavages could occur before purification steps [4,5,8,9]. Premature cleavage can be a non-negligible drawback, reducing total yields [4,9]. Therefore, several split intein purification tags have been developed for inactivating intein activity, thereby overcoming this problem [10,11]. We took a different approach by developing a salt-inducible-intein system, in which the salinity condition could control cleavage/protein-splicing reactions, thereby suppressing premature cleavages completely (Figure 1b) [12]. Many inteins from extremely halophilic archaea are seemingly incapable of protein splicing under low salinity [12,13,14]. Conditional protein splicing (CPS) by adjusting salt conditions could be a method controlling protein-splicing/cleavage reactions without any background activities observed for other CPS strategies [12,14,15]. Thus, salt-inducible inteins could efficiently circumvent the premature cleavage problem observed in the IMPACT™ system without splitting inteins for the inactivation. CPS by salts was indeed successfully demonstrated for protein ligation by protein trans-splicing [12,16].

In this work, we developed a novel salt-cleavable tag, termed the Inducible intein-mediated self-cleaving tag (IIST), derived from MCM2 intein from the extremely halophilic archaeon, *Halorhabdus utahensis* (*Hut*MCM2 intein). We also demonstrated the amidation of a V_HH_ domain using the IIST system by incorporating ^15^N atoms at the C-terminus of the ^15^N-labeled ammonium salts supplied for inducing the cleavage. Finally, we also exemplified the applications of the IIST system with small peptides.

## 2. Results

### 2.1. Design of the Minimized HutMCM2 Intein

We previously reported protein purification using salt-inducible inteins from extremely halophilic organisms [12]. We first analyzed the salt-dependent protein-splicing mechanism of the MCM2 intein from *Halorhabdus utahensis* (*Hut*MCM2) by NMR spectroscopy. The [^1^H,^15^N]-HSQC spectrum of the C1A variant of the *Hut*MCM2 intein without any salt showed highly degenerated peaks around 8.5 p.p.m. at the ^1^H frequency, indicating that the *Hut*MCM2 intein is highly disordered without any defined three-dimensional structure (Figure 2a). In contrast, the [^1^H,^15^N]-HSQC spectrum in the presence of 3.4 M NaCl showed well-dispersed signals indicating a well-defined three-dimensional structure (Figure 2b). These NMR spectra revealed that the *Hut*MCM2 intein was inactive due to the absence of an active three-dimensional structure, which could be induced by a high salt concentration such as 3 M NaCl. Thus, premature cleavages of the *Hut*MCM2 intein could be effectively suppressed by the lack of active conformations without lowering the expression temperature used for the IMPACT™ system [3,4]. The *Hut*MCM2 intein consists of 186 residues, which is relatively large compared with other mini-inteins such as a *cis*-splicing variant of *Npu*DnaE intein bearing only 136 residues [17]. Because a smaller purification tag is usually preferable for higher yields due to the relative molecular mass, we decided to minimize the *Hut*MCM2 intein. Based on the backbone resonance assignments of the C1A variant of the *Hut*MCM2 intein (unpublished), we deleted 20 residues corresponding to a presumable loop region of the *Hut*MCM2 intein judging from the degenerated NMR signals (Figure 2c). To check the salt-dependency of the minimized *Hut*MCM2 intein (*Hut*MCM2^Δ20^), we recorded the [^1^H,^15^N]-HSQC spectrum of the C1A variant of the *Hut*MCM2^Δ20^ intein in the presence of a high salt concentration, which was very similar to the one from the original C1A variant of the *Hut*MCM2 intein with well-dispersed peaks (Figure 2d). This data indicated that the 20-residue deletion did not affect the folding of the *Hut*MCM2^Δ20^ intein in the presence of a high salt concentration. These NMR data were in line with the splicing activity of the *Hut*MCM2^Δ20^ intein under 3 M NaCl (Figure 2e). Thus, we used the *Hut*MCM2^Δ20^ intein to develop a salt-cleavable intein-mediated purification system.

### 2.2. Purification and the C-Terminal Amidation of a Single Domain Antibody (V_HH_) with IIST

Next, we tested purification and amidation using a small model protein, a single VH-like domain (V_HH_) (a single domain antibody, or a nanobody) from llama [18,19]. The V_HH_ domain against the human chorionic gonadotropin hormone (hCG) alpha subunit (V_HH_-H14) was fused at the *N*-terminus of the codon-optimized *Hut*MCM2^Δ20^ intein with a three-residue linker (GSR), followed by a chitin-binding domain (CBD) and an octahistidine tag (Figure 3a). We used an octahistidine-tag because halophilic proteins like *Hut*MCM2 intein are very acidic, reducing the binding to an IMAC column [20,21,22]. We also included a CBD domain as an alternative binding domain because a poly-histidine tag might contaminate heavy metals in the final product, making it less favorable with biopharmaceuticals. The total length of IIST using the *Hut*MCM2^Δ20^ intein was still 237 residues. We also incorporated *MalE* signal peptide because a signal peptide is frequently required to produce correctly folded V_HH_ in *E. coli.* The tagged protein was expressed in *E. coli* and purified using immobilized metal affinity chromatography (IMAC). First, we cleaved the C-terminally tagged V_HH_-H14 under various conditions (Figure 3b). A reducing agent such as TCEP or DTT was required for cleavage with 2 M NaCl. For the amidation using IIST, the elution fraction from the first IMAC was concentrated and diluted into a final concentration of 1 M (^15^NH_4_)_2_SO_4_, 2 M NaCl, 50 mM sodium phosphate, and 50 mM DTT, at pH 7.0, and incubated overnight. The cleaved and modified V_HH_-H14 was collected as flow-through fractions from the IMAC and concentrated (Figure 3c). Unlike cleavage by a reducing agent, we observed two bands for V_HH_-H14 (Figure 3c). This observation suggests that about 50% of V_HH_ was amidated. This estimation is in line with the previous report in which 3 M (NH_4_)_2_CO_3_ was required for achieving >90% amidation [8]. The ^15^N-labeled ammonium carbonate would be more suitable for the selective ^15^N-labeling of the C-terminal amide group because ammonium sulfate is commonly used for precipitating proteins.

Since the theoretical mass difference between the C-terminal carboxyl and amide groups is only 1 Dalton, we used NMR spectroscopy to illustrate the amidation by introducing a ^15^N atom at the C-terminal amide group of the provided ammonium salts (Figure 3d). The [^1^H,^15^N]-HSQC spectrum of the cleaved V_HH_-H14 using IIST in the presence of ^15^N-labeled ammonium sulfate showed only two correlation peaks between two amide protons and one ^15^N atom, confirming that the amidation was introduced from the supplied ^15^N ammonium sulfate (Figure 3d). Furthermore, isotope shifts for the ^15^N atom due to 10% D_2_O in the solution were visible as additional small peaks (Figure 3d).

### 2.3. Comparison of Cleavage Conditions with a Model Protein, GFP

A high salt concentration of 3–4 M NaCl for conditional protein splicing might not be ideal for many target proteins. Therefore, we were also interested in lowering the salt concentration for the inducible-intein cleavage. We tested various cleavage conditions using a model target protein, a green fluorescent protein (GFP) (Figure 4a) [12]. We observed similar cleavage efficiencies for 2–4 M NaCl using the model protein (Figure 4b,c). Interestingly, we noticed that sodium phosphate requires less concentrations than sodium chloride for the cleavage. 0.5 M sodium phosphate (NaPi) reached >50% cleavage after 20-h incubation (Figure 4c). The NMR study indicated the equilibrium between the folded and unfolded states of the *Hut*MCM2 intein, of which population could be shifted by varying the salt concentration. To stabilize the active folded state of the *Hut*MCM2 intein, we tested a stabilizing co-solvent of sucrose instead of salts [23]. Indeed, 0.5–1 M sucrose, which is known to stabilize proteins by preferential hydration [23], increased the cleavage efficiency together with other salts such as 0.5 M sodium phosphate (Figure 4d). This observation suggests that other co-solvents stabilizing the active protein structure could also induce self-cleavages by the *Hut*MCM2 intein instead of a high concentration of salts.

### 2.4. Applications of IIST with Small Peptides

Next, we tested the IIST system for the purification of small peptides. For this purpose, we selected several bioactive peptides with less than 50 residues, to which carboxyl-terminal amidation could be beneficial. We chose human growth hormone-releasing factor (hGHRF, 44 amino acids) [24], sarcotoxin IA (39 amino acids) [25], exenatide (glucagon-like peptide-1 receptor agonist, 39 amino acids) [26], and ω-conotoxin MVIIC (26 amino acids) [27] (Figure 5a). The yields were between ca. 5–20 mg/L for these peptides after the first IMAC purification (Figure 5b). The worst yield of ca. 5 mg/L was obtained for sarcotoxin IA, possibly due to the antibacterial activity. Sarcotoxin IA fusion was mostly degraded during the salt incubation. We obtained ca. 16 mg/L for hGHRF-IIST after the first IMAC. Despite the absence of a high salinity condition, we still observed presumable premature cleavage, thereby significantly reducing the yield (Figure 5c). The cleaved hGHRF was hardly visible on the tricine SDS-PAGE. The yield for ω-conotoxin MVIIC-IIST was estimated to be ca. 11 mg/L. The induced cleavage for ω-conotoxin MVIIC-IIST was the worst of about 30% because of the amino-acid type at the cleavage site (Figure 5d). Therefore, we did not characterize these peptides any further. Unlike other peptides, we obtained about 20 mg/L of the tagged exenatide before removing IIST, corresponding to the maximum theoretical yield of about 3 mg/L. We cleaved the tagged exenatide with 0.5 M NaCl, 0.5 M sodium phosphate, pH 7, and 0.5 mM TCEP for 2-day incubation, followed by column chromatography using chitin resins to remove IIST (Figure 5e). Interestingly, the higher concentration of 2 M NaCl showed worse cleavage than the 0.5 M NaCl for exenatide-IIST. Despite the long incubation with the optimized salt condition, the cleavage of the tag was incomplete, suggesting that the junction amino-acid type of “Ser” might influence cleavage efficiency as reported for the IMPACT™ system (Figure 5e) [5,28,29].

### 2.5. Optimization of IIST

The yields per liter using the first version of IIST were not as high as we anticipated. Therefore, we decided to remove the CBD, although a poly-histidine tag might not be suitable to produce biopharmaceuticals. The removal of CBD reduced the size of IIST to 187 residues (Figure 6a). We could typically obtain about 50 mg/L for the fusion protein bearing a glucagon-like peptide-1 (GLP-1) analog (31 amino acids) and about 52 mg/L for ω-conotoxin MVIIC-IIST construct. The yield was more than twice that of the original IIST (Figure 6b). However, the salt-cleavage of ω-conotoxin MVIIC-IIST was inefficient at <25%, presumably due to the Cys residue of ω-conotoxin MVIIC at the cleavage site. Thus, we could not observe enough ω-conotoxin MVIIC on the tricine SDS-PAGE. Therefore, we used a GLP-1 analog instead of the original exenatide because Arg residue immediately upstream of IIST could be used. The *Hut*MCM2 intein has the wild-type Arg at the so-called -1 position [12,16]. The protein-splicing activity of the *Hut*MCM2 intein was the best when the -1 position was Arg in the model system [29]. We estimated the cleavage efficiency of the GLP-1 analog to be >80% after 18 h (Figure 6b). We obtained ca. 3–4 mg/L after removing IIST, which was about 40–50% of the theoretical maximum yield (Figure 6c). We tested the amidation of a GLP-1 analog in 10 mM TCEP, 50 mM sodium phosphate, pH 7, 1 M (NH_4_)_2_SO4, and 2 M NaCl at 25 °C for 20 h and analyzed the purified peptide after the second IMAC using reverse-phase liquid chromatography and MALDI-TOF mass spectrometry (Appendix A).

## 3. Discussion

An efficient purification method for recombinant peptides and proteins is highly desirable. Notably, many bioactive peptides are amidated at the carboxyl ends for their complete activities. Therefore, we developed the IIST system utilizing the salt-dependent *Hut*MCM2 intein. Importantly, IIST can be cleaved off by a high salt concentration under a reducing condition without any proteolytic enzyme. The salt concentration could be reduced to 0.5 M sodium phosphate when co-solvent such as sucrose was used. The *HutMCM2* intein is unfolded and inactive without any salts as evidenced with the NMR spectra, thereby suppressing undesired cleavages before the purification steps, unlike the inteins used in the IMPACT™ system [5]. The equilibrium between the unfolded and folded states of the *HutMCM2* intein can be controlled by adjusting the salt concentration. We demonstrated that co-solvent sucrose stabilizing the folded state could also induce the cleavage reaction, suggesting that other co-solvents known for stabilizing protein structures might also be used for cleaving IIST without any salts [31].

As previously demonstrated with the IMPACT™ system [8], ammonium ions could function as nucleophiles to cleave the thioester intermediate, resulting in amidation (Figure 1). Therefore, the IIST system could also be applicable for the C-terminal amidation of *N*-terminally fused targets by adding a high concentration of ammonium salts during the cleavage reaction [8]. We visualized the amidation in the HSQC spectrum using ^15^N-labeled ammonium sulfate. Other ^15^N-labeled ammonium salts such as ammonium carbonate might be optimal for using a higher concentration of ammonium salts and could result in a higher amidation [8]. ^15^N-labeling at the C-terminal ends of proteins and peptides using the IIST system could provide a new tool for various NMR studies, such as monitoring structural changes near the C-terminus of proteins.

We tested the IIST system with several peptides. Their yields were disappointingly modest for the tagged proteins (5–20 mg/L). Shortening the IIST by removing CBD increased the yield twice or more. After optimizing the IIST, the best yield (ca. 3–4 mg/L) was obtained for the GLP-1 analog after removing IIST. Even though the IIST system generally does not suffer from the premature cleavages observed with the IMPACT™ system, we experienced poor cleavages of IIST for some peptides. This observation indicated that the amino-acid type upstream of the IIST could affect the cleavage efficiency, similar to protein splicing [3,4,5,28,29].

Moreover, the optimized IIST is still as large as 20-kDa, although the 20-residue deletion contributed a ca. 10% reduction to the molecular mass of IIST. The 10% smaller size for IIST could improve the final theoretical yield by a few percent depending on the target size. A 5-kDa peptide would theoretically result in only up to 20% of the total yield with a 20-kDa IIST, even at 100% cleavage efficiency. Therefore, the initial yield of a tagged protein dominantly determines the final yield after removing IIST, considering the observed initial yields (5–20 mg/L). When we removed CBD from IIST, 50 mg/L of the tagged protein was achieved for a GLP-1 analog, resulting in a final yield of 3–4 mg/L. The critical limiting steps of the IIST system might be the optimization of the protein expression and cleavage conditions. We also noticed that the yield and the optimal salt condition for cleavages depended on the target peptide. The prior information on cleavage efficiencies for the 20 different amino-acid types of *Hut*MCM2 intein at the cleavage junction might be advantageous for applying the IIST system [29].

One way to make the IIST system more robust might be applying a smaller, more robust intein in the IIST system because of the molecular mass effect. For example, the theoretical yield with a 10-kDa IIST, instead of a 20-kDa IIST, would improve by 13% for a 5-kDa peptide. Several inteins are indeed smaller than 130 residues among >1500 identified inteins, although they are probably not strictly halophilic inteins [32]. Thus, the biochemical characterization of inteins from various halophilic organisms might contribute to developing a smaller and better self-cleaving tag for biotechnological applications.

## 4. Materials and Methods

### 4.1. Constructions of Plasmids for Protein Expression

The full-length *Hut*MCM2 intein with C1A mutation used for NMR studies was cloned into a vector pHYRSF53 [33] between the *Bam*HI and *Hind*III sites by amplifying the gene from *Halorhabdus utahensis* genomic DNA using the two oligonucleotides I135: 5′-TCGGATCCATGCGGGCCGTTACTGGGGATACTC and I379: 5′-GTAAAGCTTAATTATGGACGACCATTCCG, resulting in the plasmid pSCFRSF125 [12,33]. Plasmid pJODuet142 bearing the *cis*-splicing precursor containing the *Hut*MCM2^Δ20^ intein with two GB1s as exteins (H_6_-GB1-*Hut*MCM2^Δ20^-GB1), was constructed from pSADuet616 by inverse PCR using two oligonucleotides, I528: 5′-GTCCCGGAAGGCCCGGCGGAATCCGGACTCG and I529: 5′-CGGATTCCGCCGGGCCTTCCGGGACGAAAAC [12]. The *Hut*MCM2^Δ20^ intein with the C1A mutation for the NMR studies was derived from pJODuet142 using two primers of I135 and I379, resulting in pBHRSF128. The genes for the anti-human chorionic gonadotropin hormone (hCG) alpha subunit V_HH_ single-domain antibody (V_HH_-H14) with the *MalE* signal peptide [18,19] and the codon-optimized *Hut*MCM2^Δ20^ intein with AA mutations together with the C-terminally octahistidine-tagged chitin-binding domain (CBD) were chemically synthesized and purchased from Integrated DNA Technologies, BVBA (Leuven, Belgium) as the plasmids pIDTSMART-KAN-GeneSyn15 and pIDTSMART-KAN-GeneSyn13, respectively. The two genes were ligated into a pRSF vector, resulting in pBHRSF99. This plasmid was used to produce V_HH_-H14 with an IIST. The gene of the codon-optimized *Hut*MCM2^Δ20^ intein with CBD from pIDTSMART-KAN-GeneSyn13 was also cloned into a pET vector (pET-GFP LIC cloning vector (2GFP-T), addgene plasmid #29716) using the two restriction sites *Ssp*I and *Hind*III, resulting in pBH(etGFP)Syn13. The synthetic gene for the codon-optimized *Hut*MCM2^Δ20^ intein contained an unexpected mutation of T174S.

The fusion protein gene bearing GFP was cut from the plasmid, pBH(etGFP)Syn13, and cloned into pRSF-1b between *Nco*I and HindIII sites. The N-terminal His-tag was genetically removed from the plasmid by PCR using two oligonucleotides I759: 5′-TAAGGAGATATACCATGGCTTCTTCTGTGAGCAAGG and I760: 5′-CCTTGCTCACAGAAGAAGCCATGGTATATCTCCTTA, resulting in pBHRSF142. This construct was used as GFP-IIST. For hGHRF and ω-conotoxin MVIIC, the genes of hGHRF and ω-conotoxin MVIIC were chemically synthesized and purchased as pIDTSMART-KAN-GeneSyn16 from Integrated DNA Technologies, BVBA (Leuven, Belgium) and cloned into pBH(etGFP)Syn13 using *Nde*I and *Spe*I restriction sites, resulting in pBH(et)93 and pBH(et)94, respectively. For the expression of sarcotoxin IA and exenatide, the genes of sarcotoxin IA and exenatide were chemically synthesized and purchased as pIDTSMART-KAN_geneSyn24 and pIDTSMART-KAN_GeneSyn22 from the same provider. These genes were cloned between the *Nde*I and *Spe*I sites of pBHRSF99, resulting in pLKRSF2 and pSARSF828, respectively. The shorter GLP-1 analog fusion protein was derived from pSARSF828 using two oligonucleotides I919: 5′-TGYGTTACTGGAGATACAC, I920, and 5′-GTGTATCTCCAGTAACACAGCGGCCATTTTTAAGCCACTC, resulting in pSARSF840. The coding region for CBD in pSARSF840 was removed by PCR using two oligonucleotides I791: 5′-AGCTCGACAAATCCTGGTGGTCACCATCATCATCATCATC, and I792: 5′-GATGATGATGATGGTGACCACCAGGATTTGTCGAGCTCG, resulting in pSARSF847. For ω-conotoxin MVIIC with the shorter IIST, the gene for ω-conotoxin MVIIC from pBH(et)94 was cloned into pSARSF847 using *Nde*I and *Spe*I sites, resulting in pSARSF851.

### 4.2. Protein Expression and Purification

All fusion proteins were produced in *E. coli* strain T7 Express (New England Biolabs) using one of the above plasmids. For 100% [^13^C,^15^N]-labeled samples, *Hut*MCM2 and *Hut*MCM2^Δ20^ inteins with the C1A mutation were produced as H_6_-SUMO fusion proteins in M9 medium supplemented with ^15^NH_4_Cl and ^13^C_6_-glucose as the sole nitrogen and carbon sources, respectively [33]. The transformed cells were grown at 37 °C and induced with a final concentration of 1 mM isopropyl-β-D-thiogalactoside (IPTG) for 4 h. The labeled proteins were purified using a 5 mL HisTrap HP column (GE Healthcare Life Sciences) as previously described, including removing the N-terminal SUMO domain [33]. The protein was dialyzed against deionized water and concentrated using Macrosep^®^ Advance Centrifugal Devices with 5K MWCO (PALL Corporation, New York, NY, USA). 3.4 mM and 2.4 mM solutions of the *Hut*MCM2^Δ20^ (C1A) and *Hut*MCM2 (C1A) inteins in 3.1 M and 3.4 M NaCl solution, respectively, were used for NMR.

Peptides with the IIST were expressed in 2-L LB media supplemented with 25 μg/ mL kanamycin (for pSARSF847, pLKRSF2, pSARSF828, and pSARSF851) or 100 μg/mL ampicillin (for pBH(et)83 and pBH(et)84). The cultures were induced with a final concentration of 1 mM isopropyl-β-D-thiogalactoside (IPTG) for 4 h or overnight (ca 18 h) at 25 °C when OD_600_ reached 0.6. The induced cells were harvested by centrifugation at 4700× *g* for 10 min, 4 °C and lysed in 20 mL buffer A (50 mM sodium phosphate, pH 8.0, 300 mM NaCl) using an EmulsiFlex-C3 homogenizer (Avestin Inc, Ottawa, Canada) at 15,000 psi, 4 °C for 10 min. Lysates were cleared by centrifugation at 38,000× *g* at 4 °C for 60 min. The fusion proteins were purified using a 5 mL HisTrap HP column (GE Healthcare Life Sciences). The elution fractions with the fusion protein were dialyzed against deionized water and concentrated using a Macrosep^®^ Advance Centrifugal Devices 3K MWCO (PALL Corporation, New York, NY, USA).

A large-scale cleavage of a GLP-1 analog with IIST was performed with 0.5 mM solution of the fusion protein, 5 mM TCEP, 50 mM Tris-HCl, pH 7, and 3 M NaCl in a total volume of 2 mL at room temperature for 18 h and used for quantification of the yield. In addition, a large-scale amidation of the GLP-1 analog with IIST was carried out with 0.25 mM of the tagged protein, 10 mM TCEP, 50 mM sodium phosphate, pH 7, 1 M (NH4)_2_SO_4_, and 2 M NaCl in a total volume of 1.4 mL at 25 °C for 20 h. For the comparison with the amidation of the GLP-1 analog using IIST, the cleavage reaction was performed with 0.2 mM fusion protein, 10 mM TCEP, 50 mM sodium phosphate, pH 8, and 3 M NaCl in a total volume of 0.9 mL at 25 °C for 20 h. Each incubated reaction mixture was loaded on a HisTrap column after three-fold dilution with buffer A, containing 30 mM imidazole. Flow-through fractions from the column were collected and lyophilized. The lyophilized sample was further analyzed and purified by reverse-phase liquid chromatography using a SOURCE 15RPC ST 4.6/100 column (GE Healthcare, USA) with a linear gradient of 30–80% from a water/0.05% TFA to an acetonitrile/0.05% TFA (Appendix A). The dominant peak was subjected to MALDI-TOF mass spectrometry (Appendix A). A large-scale cleavage of ω-conotoxin MVIIC-IIST was performed with 0.3 mM protein solution, 2 M NaCl, 50 mM DTT, 50 mM Tris-HCl, pH 7 in a total volume of 2 mL at 25 °C for 15 h.

### 4.3. Purification and the C-Terminal Amidation of V_HH_-H14

*E. coli* T7 Express cells transformed with pBHRSF99 bearing V_HH_-H14 and IIST were grown in 2-L LB-media supplemented with kanamycin (25 μg/mL) until the OD_600_ reached 0.6 at 30 °C. The cells were induced with a final concentration of 0.5 mM IPTG and continued overnight. The cells harvested by centrifugation at 6700× *g* for 10 min were resuspended in buffer A (50 mM sodium phosphate, 300 mM NaCl, pH 8.0). After cell lysis using EmulsiFlex C-5 Homogenizer (Avestin Inc, Ontario, Canada), the soluble fraction was cleared by centrifugation at 38,500× *g* at 4 °C for one hour. The filtered sample using a 0.45 μm filter was loaded on a 5 mL HisTrap HP column (GE Healthcare Life Sciences, USA), which was washed with 50 mM imidazole in buffer A and eluted with 250 mM imidazole. The elution fractions containing V_HH_-H14 with IIST were pooled and dialyzed against MilliQ water at 10 °C, then concentrated using a Vivaspin Turbo 15 (Sartorius) centrifugal concentrator with a molecular weight cut-off (MWCO) of 5000. A final concentration of 80 μM V_HH_-H14-IIST was incubated in Amidation buffer (1 M (^15^NH_4_)_2_SO_4_, 2 M NaCl, 50 mM sodium phosphate, 50 mM DTT, pH 7.0) at 25 °C for 20 h. The precipitation that appeared during the reaction was removed by centrifugation at 15,000× *g* for 5 min. The solution for the soluble fraction was exchanged with buffer A with PD-10 column (GE Healthcare Life Sciences, USA) and then loaded on a 5-mL HisTrap HP column for the second IMAC. The flow-through fractions were collected and analyzed. The sample was concentrated. The buffer solution was changed to 20 mM sodium phosphate, pH 6.0, using a Vivaspin Turbo 15 (Sartorius) centrifugal concentrator with a MWCO of 3000. A final concentration of 0.5 mM V_HH_-H14 was transferred into a Shigemi microcell (SHIGEMI CO, Tokyo, Japan). A 230 μL solution containing 23 μL D_2_O (Aldrich, 435767-25G) of 0.5 mM ^15^N-labeled V_HH_-H14 in 20 mM potassium phosphate, pH 6.0, was used for the NMR measurements.

### 4.4. NMR Spectroscopy

[^1^H, ^15^N]-HSQC spectra were recorded at 303 K on a Bruker Avance III HD 850 MHz spectrometer with a 5 mm cryoprobe head (Bruker Corp., Billerica, MA, USA). The NMR spectra were processed with TopSpin 3.2 and presented using CcpNmr 2.4.1 [34].

## 5. Patents

PCT/FI2016/050277 was filed from work reported in this article.

## Figures and Tables

**Figure 1 molecules-26-05948-f001:**
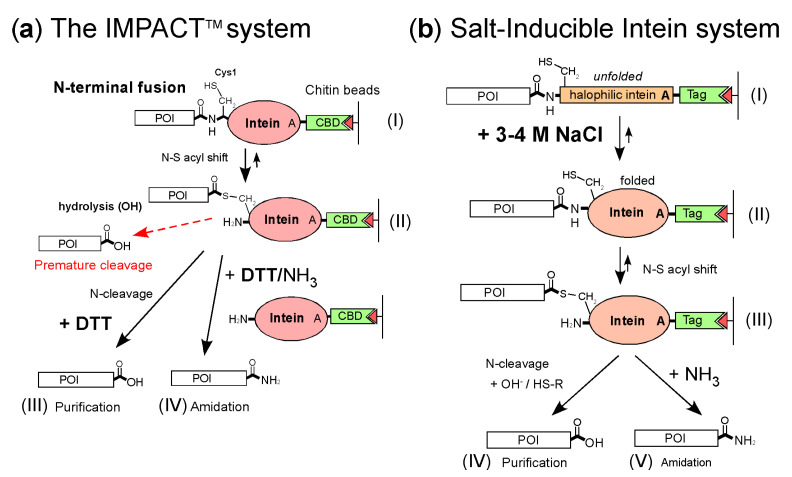
Schematic comparison between the two different purification systems (**a**) The IMPACT™ system from New England Biolabs for protein purification. The N-terminal fusion protein from a crude cell extract is purified by a chitin affinity column (I). The first N-S acyl migration of the fusion protein autocatalytically occurs and halts as the thioester intermediate due to the Ala mutation (II). The intein fusion protein is then induced to undergo on-column self-cleavage (arrow) by a reducing reagent such as dithiothreitol (DTT). The target protein is released from the column and eluted as a pure protein (III) [3,5]. The C-terminal amidation was achieved by adding DTT/ammonium ion (IV) [8]. The thioester intermediate after the first N-S acyl step could be immediately hydrolyzed, resulting in premature cleavage (a broken red arrow). (**b**) The salt-inducible intein system was developed for protein purification and amidation. The N-terminal fusion protein from a crude cell extract is purified using an affinity tag by a chitin affinity column or immobilized metal affinity chromatography (IMAC) (I). A high salt concentration of 3-4 M NaCl induces a strictly halophilic intein folding (II), and the intein undergoes an N-S acyl shift (III) [12]. The Ala mutation halts the thioester intermediate after the first N-S acyl step, cleaving through a reducing agent (IV). A high concentration of ammonium ions induces C-terminal amidation under a reducing condition using DTT (V).

**Figure 2 molecules-26-05948-f002:**
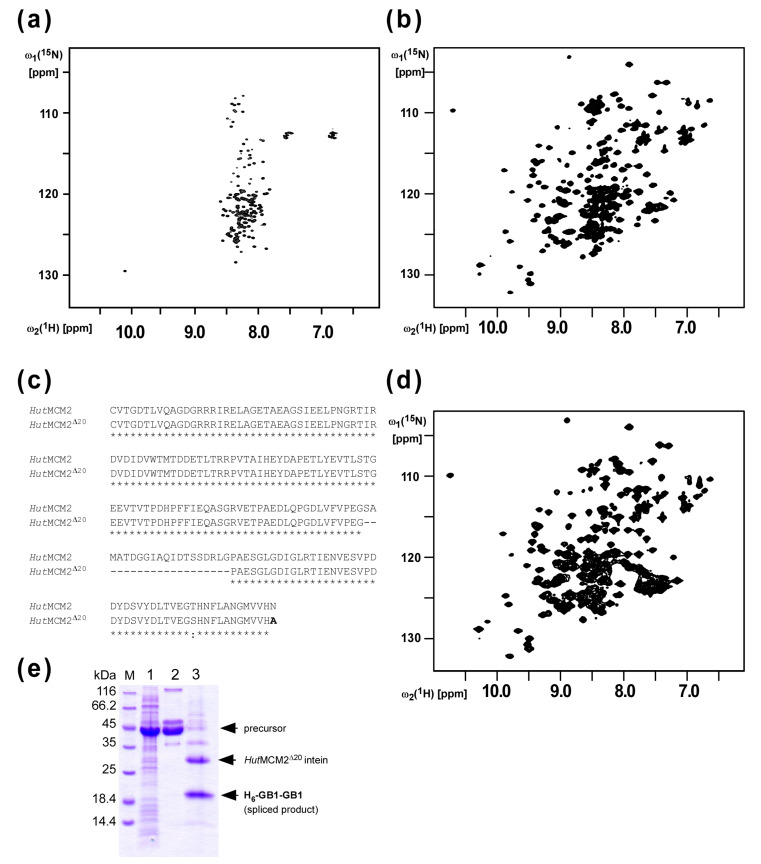
Minimization of the *Hut*MCM2 intein based on NMR analysis (**a**) The [^1^H, ^15^N]-HSQC spectrum of the *Hut*MCM2 intein without any salt and (**b**) with 3.4 M NaCl. (**c**) A sequence alignment of the wild-type *Hut*MCM2 and synthetic *Hut*MCM2^Δ20^ inteins. (**d**) The [^1^H, ^15^N]-HSQC spectrum of the *Hut*MCM2^Δ20^ intein in the presence of 3.4 M NaCl. (**e**) SDS-PAGE analysis of the *cis*-splicing of the *Hut*MCM2^Δ20^ intein. The B1 domain of IgG binding protein A (GB1) was used as the N- and C-exteins. The precursor protein (H_6_-GB1-*Hut*MCM2^Δ20^-GB1) produces the *N*-terminally his-tagged two GB1 (H_6_-GB1-GB1) and the excised *Hut*MCM2^Δ20^ intein upon protein splicing. Lane 1, total cell lysate after protein expression; lane 2, elution from IMAC; lane 3, after incubation in the presence of 2 M NaCl.

**Figure 3 molecules-26-05948-f003:**
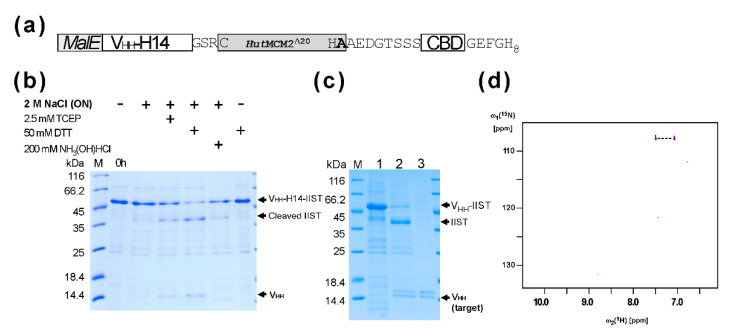
Purification and amidation of a model protein V_HH_-H14 using IIST. (**a**) The schematic structure of the fusion protein containing V_HH_-H14 with IIST for purification and amidation. (**b**) SDS-PAGE analysis of cleavages of the tagged protein under various conditions. (**c**) SDS-PAGE analysis of the amidation/cleavage and purification of V_HH_-H14. M stands for molecular weight markers. Lane 1, elution fraction from IMAC; lane 2, after incubation in the presence of the salt mixture; lane 3, the flow-through fraction from IMAC after salt incubation. (**d**) The [^1^H, ^15^N]-HSQC spectra of the cleaved and purified V_HH_-H14 after salt incubation using ^15^N-labeled ammonium sulfate during the cleavage reaction. The dotted line connects two protons attached one ^15^N atom in the C-terminal amide group.

**Figure 4 molecules-26-05948-f004:**
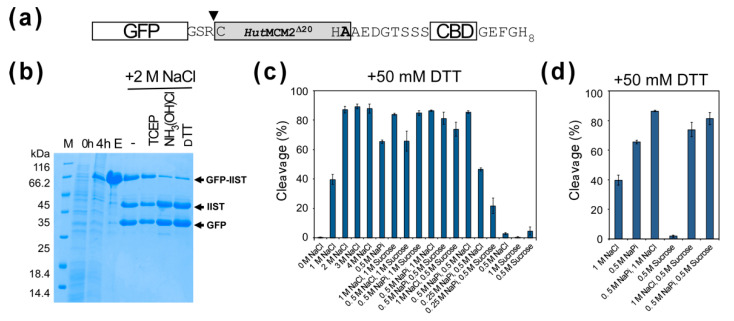
Cleavage tests of GFP-IIST with various conditions. (**a**) A schematic drawing of the construct using GFP with IIST. Only three amino-acid residues between GFP and IIST are shown at the junction. An inverse triangle indicates the cleavage site. (**b**) SDS-PAGE analysis of cleavages of GFP with IIST under 2 M NaCl using different cleavage conditions. M stands for molecular weight markers. Lanes 0 h, 4 h, and E stand for before induction, 4 h after protein expression, and the elution fraction from IMAC, respectively. “-“, TCEP, NH_3_(OH)Cl, and DTT indicate no additive, 0.5 mM tris(2-carboxyethyl)phosphine (TCEP), 100 mM hydroxylamine, and 50 mM DTT, respectively. (**c**) A plot of cleavage efficiencies under various conditions using GFP-IIST. (**d**) A plot shows the additive effects of the co-solvents.

**Figure 5 molecules-26-05948-f005:**
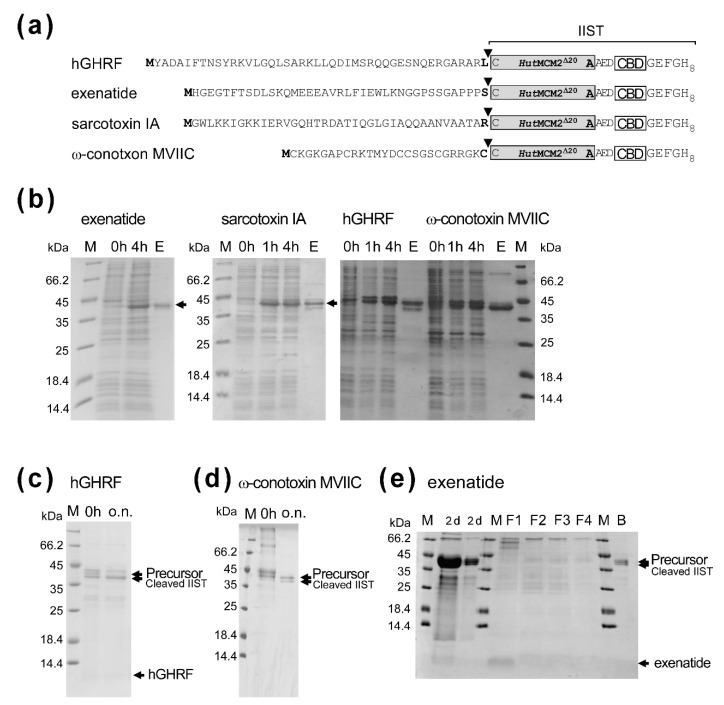
Applications of IIST to small peptides. (**a**) Schematic drawings of fusion constructs with various peptides. Inverse triangles indicate the cleavage sites. (**b**) SDS-PAGE analysis of protein expression and purification with exenatide, sarcotoxin IA, hGHRF, and ω-conotoxin MVIIC. M stands for molecular weight markers. 0 h, 1 h, 4 h, and E stand for samples at 0 h, 1 h, 4 h after protein expression, and elution from IMAC, respectively. Unlabeled arrows indicate bands for fused precursor proteins below the 45-kDa marker. (**c**) Salt-cleavage of hGHRF-IIST. The purified hGHRF-IIST (precursor) by IMAC was incubated in the salt solution (50 mM DTT, 2 M NaCl) overnight (ca. 18 h). (**d**) Salt-cleavage of ω-conotoxin MVIIC-IIST. The purified ω-conotoxin MVIIC-IIST (precursor) was cleaved in the salt solution (10 mM DTT, 0.5 M sodium phosphate, 0.5 M NaCl). 0 h and “o.n.” indicate the samples before incubation and after 20-h incubation, respectively. (**e**) Tricine-SDS-PAGE analysis of the purification and cleavage of the tagged exenatide. M stands for molecular weight markers. “2 d” indicates samples after a 2-day cleavage reaction. F1, F2, F3, and F4 indicate the flow-through fractions from the chitin-beads column. B stands for the bound fraction to chitin beads. 20% Tricine-SDS-PAGE was used to visualize exenatide [30].

**Figure 6 molecules-26-05948-f006:**
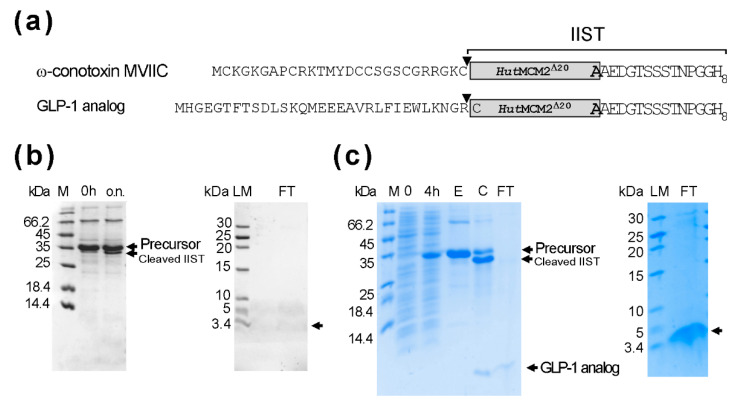
Expression and purification of a GLP-1 analog and ω-conotoxin MVIIC using the smaller IIST (**a**) A schematic drawing of the fusion constructs bearing a GLP-1 analog or ω-conotoxin MVIIC. Inverse triangles indicate the cleavage sites. (**b**) Salt-cleavage of ω-conotoxin MVIIC with the smaller IIST. 0 h and “o.n.” indicate the samples before the salt addition or after 15-h incubation in the salt solution (2 M NaCl, 50 mM DTT, 50 mM Tris-HCl, pH 7). LM and FT indicate low molecular weight markers and the flow-through fraction from the second IMAC, respectively. (**c**) Expression, purification, and cleavage of a GLP-1 analog with IIST. M stands for molecular weight markers. Lanes 0 h, 4 h, and E stand for samples at 0 h and 4 h after the protein expression and elution from the first IMAC, respectively. C and FT stand for the samples taken from the cleavage reaction and the flow-through fraction from the second IMAC, respectively. The samples were analyzed by 20% tricine-SDS-PAGE and stained with Coomassie brilliant blue. Tricine-SDS-PAGE analysis of the purified peptide from the second IMAC [30]. LM and FT indicate low molecular weight markers and the flow-through fraction from the second IMAC, respectively. Unlabeled arrows indicate the expected positions for the bands of the peptides.

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
