# Peer review of "The Inducible Intein-Mediated Self-Cleaving Tag (IIST) System: A Novel Purification and Amidation System for Peptides and Proteins"

_molecules, 2021, doi:10.3390/molecules26195948_

Round 1
Reviewer 1 Report
The manuscript by Aranko and Iwai describes the characterization of a new Intein-mediated protein cleavage that is specifically induced by high salt addition. Although some parts of the finding are interesting (e.g. the production of small peptides and C-terminal amidation of peptides), I am not fully convinced that this system (the SCIMP) would be useful for recombinant protein production. Using high salt concentration (3-4 M NaCl) often ruins the function of proteins. In order to claim that the SCIMP system is suitable for recombinant protein production, more than 2 model proteins should be tested, including enzymes.
My major concerns:
- The manuscript is written in poor English, therefore it must be carefully rewritten and proofread before re-submission.
- Not only the grammar, but also phrasing need to be improved and in some cases explained:
Lane 14: “have often” should be “have been”
Lane 18: “the halophilic organism” should be “a halophilic organism” or “the halophilic organism, xyz”.
Lane 1-20: “salt-inducible intein by demonstrating selective…” should be “salt-inducible intein demonstrated by selective”. This appears elsewhere…
Lane 29: there is no such a thing that “recombinantly produced”. Only the protein (or the encoding DNA) can be recombinant. Correctly: “recombinant proteins and peptides produced in…”. This problem appears elsewhere…
Lane 34: “as a kit” is not needed
Lane 59: “halophilic archeon”, again the reader does not have a clue which organism is this.
Lane 66: “Biolab” is “Biolabs”
Lane 80: Figure 1a is not explained in the text.
Lane 90: why these 20 residues were selected and deleted? Please explain.
Lane100: Figure 2e: please explain the lanes (1-3) in the figure legend. What is lane 1, 2 and 3?
Lane 105: “H6-GB1-GB1” this protein appeared from nowhere. It was not explained in the text nor in the figure legend. What is this protein and what exactly do we see in the gels?
Lane 106: “the” is not needed. Since SCIMP as a mosaic word was already introduced in the text, there is no need to include both “Salt-Cleavable Intein-Mediated Purification” and “SCIMP” in this sub-tilte.
Lane 106: I think this sub-tile is not appropriate. The title of Figure 3 (Purification and amidation of a model protein VHH-H14 using the SCIMP-tag) would be better.
Lane 111: Why did the authors use two tags (CBD and His8)? If HisTrap was used to purify the proteins, the CBD (suitable for chitin-resin-purification) was unnecessary. Please explain.
Lane 115: “at the front” is scientifically incorrect. It should be “signal peptide fused to the N-terminus of...”.
Lane 119: “overnight” is inappropriate. Please indicate the incubation time exactly.
Lane 143-144: “The yields from the first IMAC were between 5-20 mg/L for these peptides after IMAC purification using the Ni-NTA column”. There are two problems here: IMAC (Immobilized metal affinity chromatography) as a methos is basically the same as Ni-NTA (please rephrase the sentence); second, the authors claim in the M&M that they used HisTrap HP columns from GE Healthcare, which is not Ni-NTA.
Lane 256/286: either E. coli or Escherichia coli.
- SCIMP: I do not think SCIMP is the best name for this system. First of all, because the name SCIMP is already occupied (https://www.nature.com/articles/ncomms14133). Second, practically the SCIMP method described here is not a Salt-Cleavable Intein-Mediated Purification system. It is rather a Salt-inducible Intein-mediated purification/cleavage system.
Author Response
Responses to reviewer 1
Comments and Suggestions for Authors
The manuscript by Aranko and Iwai describes the characterization of a new Intein-mediated protein cleavage that is specifically induced by high salt addition. Although some parts of the finding are interesting (e.g. the production of small peptides and C-terminal amidation of peptides), I am not fully convinced that this system (the SCIMP) would be useful for recombinant protein production. Using high salt concentration (3-4 M NaCl) often ruins the function of proteins. In order to claim that the SCIMP system is suitable for recombinant protein production, more than 2 model proteins should be tested, including enzymes.
Reponse: We have not claimed that the system is generally suitable for the production of all proteins. We rather emphasized applications for smaller proteins and peptides as shown with VHH and peptides of <50aa because they are usually unaffected by a higher salt condition. It is noteworthy that, to our surprise, a few proteins we tested are tolerant of 3-4 M NaCl (data not included). We emphasized the application to peptides and small proteins in the revised manuscript. Moreover, we have a follow-up study to lower 3-4 M Cl to 1 M NaCl for the same purpose, which was not included because it is out of the scope of this article.
My major concerns:
The manuscript is written in poor English, therefore it must be carefully rewritten and proofread before re-submission.
Lane 14: “have often” should be “have been”
Response: We corrected.
Lane 18: “the halophilic organism” should be “a halophilic organism” or “the halophilic organism, xyz”.
Response: corrected.
Lane 1-20: “salt-inducible intein by demonstrating selective…” should be “salt-inducible intein demonstrated by selective”. This appears elsewhere…
Response: We rephrased.
Lane 29: there is no such a thing that “recombinantly produced”. Only the protein (or the encoding DNA) can be recombinant. Correctly: “recombinant proteins and peptides produced in…”. This problem appears elsewhere…
Response: we rephrased.
Lane 34: “as a kit” is not needed
Response: changed.
Lane 59: “halophilic archeon”, again the reader does not have a clue which organism is this.
Response: we changed
Lane 66: “Biolab” is “Biolabs”
Response: corrected.
Lane 80: Figure 1a is not explained in the text.
Response: We added more extensive explanations in Figure 1.
Lane 90: why these 20 residues were selected and deleted? Please explain.
Response: We explained in Line 89-90 in the previous manuscript. We did NMR backbone assignments of HutMCM2 intein and deleted them based on the NMR resonance assignment. We explained this further in the text.
Lane100: Figure 2e: please explain the lanes (1-3) in the figure legend. What is lane 1, 2 and 3?
Response: We added them.
Lane 105: “H6-GB1-GB1” this protein appeared from nowhere. It was not explained in the text nor in the figure legend. What is this protein and what exactly do we see in the gels?
Response: It was clearly described in Line 104-105 in the previous version as “The B1 domain of IgG binding protein A (GB1) was used as N- and C-exteins. The precursor protein produces the N-terminally his-tagged two GB1 (H6-GB1-GB1)”. We now wrote them in the caption as well.
Lane 106: “the” is not needed. Since SCIMP as a mosaic word was already introduced in the text, there is no need to include both “Salt-Cleavable Intein-Mediated Purification” and “SCIMP” in this sub-tilte.
Response: We changed the name, considering suggestions by the reviewers.
Lane 106: I think this sub-tile is not appropriate. The title of Figure 3 (Purification and amidation of a model protein VHH-H14 using the SCIMP-tag) would be better.
Response: We changed.
Lane 111: Why did the authors use two tags (CBD and His8)? If HisTrap was used to purify the proteins, the CBD (suitable for chitin-resin-purification) was unnecessary. Please explain.
Response: We originally wanted to use only CBD because a His-tag is not suitable for GMP production of biopharmaceuticals due to a heavy metal used in IMAC. For laboratory use, His-tag is more robust and straightforward for purification. Therefore, we incorporated both initially.
Lane 115: “at the front” is scientifically incorrect. It should be “signal peptide fused to the N-terminus of...”.
Response: we changed.
Lane 119: “overnight” is inappropriate. Please indicate the incubation time exactly.
Response: we changed.
Lane 143-144: “The yields from the first IMAC were between 5-20 mg/L for these peptides after IMAC purification using the Ni-NTA column”. There are two problems here: IMAC (Immobilized metal affinity chromatography) as a methos is basically the same as Ni-NTA (please rephrase the sentence); second, the authors claim in the M&M that they used HisTrap HP columns from GE Healthcare, which is not Ni-NTA.
Response: we changed.
Lane 256/286: either E. coli or Escherichia coli.
Response: we changed.
SCIMP: I do not think SCIMP is the best name for this system. First of all, because the name SCIMP is already occupied (https://www.nature.com/articles/ncomms14133). Second, practically the SCIMP method described here is not a Salt-Cleavable Intein-Mediated Purification system. It is rather a Salt-inducible Intein-mediated purification/cleavage system.
Response: We appreciate the suggestion. “SCIMP”, pointed out by the reviewer, is a name for the protein described in the cited article. We believe that both names will not be easily mixed up.
It is not uncommon to have the same name for different usages. For example, “IMPACT” used by NEB is also used for a protein called “IMPACT”, (imprinted and ancient gene protein homolog, https://www.uniprot.org/uniprot/Q9P2X3). Therefore, we decided to combine “Tag” and use “SCIMPT”.
Reviewer 2 Report
The authors describe further development of the salt-inducible self-cleavable purification tag based on the intein from halophilic bacterium Halorhabdus utahensis, as the extension of their previous work published in 2016 in JMB (A. Ciragan et al., J. Mol. Biol. (2016) 428:4573-4588, 10.1016/j.jmb.2016.10.006). Here, the authors report slightly truncated variant of the original intein-based tag. Therefore, submitted manuscript reports only slight and modest tweaking of already published self-cleavable purification tag and has low scientific significance.
Furthermore, the presented research has serious methodological flaws. Many results presented, such as utility of described self-cleavable tag in amidation or bioactive peptide production are not sufficiently and/or convincingly documented.
Overall, the manuscript in present form does not merit to be published and should be rejected due to lack of scientific novelty and poor methodological quality.
Major concerns:
- Figure 2b, SDS-PAGE analysis of cleavage and purification of VHH-H14: The desired cleavage product of the model protein consists of two bands. This is highly undesirable because it points to sample heterogenity. Explanation and/or identification of observed products should be provided
- H.One of the main points of the submitted manuscript is application of the SCIMP-tag to small peptides production (section 2.3 and 2.4) This part of the research is particularly ill presented and unconvincing, because desired products, bioactive peptides, are visualized and analyzed inadequate method - SDS-PAGE. SDS-PAGE is unsuitable for analysis of short peptides (peptides of MW less than 10 kDa), as evident on Figure 5. Instead, purity and identity of target peptides should be assayed by MALDI or ESI-MS, or Tricine SDS-PAGE (see: H. Schägger, Nat Protoc (2006) 1:16–22. https://doi.org/10.1038/nprot.2006.4, and Haider S. R. et al. Methods Mol. Biol. (2019) 1855:151-160.
doi: 10.1007/978-1-4939-8793-1_15) - Amidation of a model protein VHH-H14 using the SCIMP-tag is also inadequatly presented, by single evidence, [1H, 15N]-HSQC
spectrum (Figure 3c). From the 2D-NMR spectrum it is impossible to judge yield and extent of target protein amidation. Please provide more data about efficiency and yield of amidation.
Minor comments:
- The title should be corrected, it contains a typo: Intein-Medicated to Intein-Mediated
- Please explain or comment what is the reason for intein S to T mutation, close to C-terminus, as shown in Figure 2c.
Round 2
Reviewer 1 Report
My opinion is that the manuscript is acceptable for publication. I have to acknowledge, that authors did a very good job with the rewriting and proofreading of the manuscript (track changes are still visible). And I also accept almost all answers they provided. However, I still do not think that the system they developed and want to publish is:
- good for most (especially GMP) proteins (even if salt is decreased to 1M)
- the tile is still misleading: their system is NOT a "salt-cleavable" but "salt-inducible" "intein-mediated cleavage....". These are two different things.
- His-tag (IMAC Purification) is absolutely suitable for GMP production. This conclusion is wrong.
Author Response
Responses to reviewer 1
“My opinion is that the manuscript is acceptable for publication. I have to acknowledge, that authors did a very good job with the rewriting and proofreading of the manuscript (track changes are still visible). And I also accept almost all answers they provided. However, I still do not think that the system they developed and want to publish is:
- good for most (especially GMP) proteins (even if salt is decreased to 1M)
- the tile is still misleading: their system is NOT a "salt-cleavable" but "salt-inducible" "intein-mediated cleavage....". These are two different things.
- His-tag (IMAC Purification) is absolutely suitable for GMP production. This conclusion is wrong.”
“good for most (especially GMP) proteins (even if salt is decreased to 1M)”
Response:
We do not use the word of GMP in our manuscript but “biopharmaceuticals”. His tag is not the first choice for the purification of biological drugs. We also added a new section concerning the salt concentration (section 2.3 in the revised manuscript). In principle, it does not have to be even salt, but any co-solvents stabilizing the intein structure could be used, such as sucrose (included) or others (amino-acids, glycerol, etc). Therefore, we removed “salt” from the title too.
“the tile is still misleading: their system is NOT a "salt-cleavable" but "salt-inducible" "intein-mediated cleavage....". These are two different things.”
Response: We appreciate this comment and agree with the comment. Now we revised the title.
Reviewer 2 Report
The reviewer appreciates substantial improvements made to the manuscript, the authors convincingly addressed most of the reviewer's concerns.
The reviewer welcomes inclusion of the additional panel b in Figure 3 of the revised manuscript. Additional panel 3B and explanations added to the main text adequately address reviewer major concerns 1 and 3 about amidation of the model protein VHH-H14 and sample heterogenity.
Major concern:
Sections 2.3 and 2.4 describing the application of the intein-based tag to peptide production are still somewhat problematic and of low quality. The reviewer welcomes the revisions made to this part of the manuscript and inclusion of supplementary information on exenatide, a glucagon-like peptide-1 receptor agonist, but the applicability of SCIMPT system for the production of small peptides remains insufficiently supported by the experimental data. The authors have chosen 4 small peptides as the model peptides: Growth Hormone-Releasing Factor (hGHRF), sarcotoxin IA, exenatide (glucagon-like peptide-1 receptor agonist) and ω-conotoxin MVIIC. They have shown successful cleavage and production of only one (out of four) chosen peptides. They have explained the reasons why the expression and cleavage of sarcotoxin IA was unsuccessful, but they provide no data or explanation for other small peptides.
Minor concerns:
- The authors should consistently use one of the alternative names for exenatide, a glucagon-like peptide-1 receptor agonist through the text. In the section 2.3 and Figure 4 they prefer the name exenatide, while in section 2.4 and Figure 5 they use the name 'GLP-1 analog'. Consistent name usage would be preferred.
- Figure 4, panel a contains a typo: 'Exanatide' instead of 'exenatide'.
- Figure legends of Supplemental Figures should be more detailed, better described.
Author Response
Responses to Reviewer 2:
Major concern: Sections 2.3 and 2.4 describing the application of the intein-based tag to peptide production are still somewhat problematic and of low quality. The reviewer welcomes the revisions made to this part of the manuscript and inclusion of supplementary information on exenatide, a glucagon-like peptide-1 receptor agonist, but the applicability of SCIMPT system for the production of small peptides remains insufficiently supported by the experimental data. The authors have chosen 4 small peptides as the model peptides: Growth Hormone-Releasing Factor (hGHRF), sarcotoxin IA, exenatide (glucagon-like peptide-1 receptor agonist) and ω-conotoxin MVIIC. They have shown successful cleavage and production of only one (out of four) chosen peptides. They have explained the reasons why the expression and cleavage of sarcotoxin IA was unsuccessful, but they provide no data or explanation for other small peptides.
Response: We added more data on the cleavages of hGHRF and conotoxin (section 2.4). Both were not very successful for further characterizations. We discussed this in the revised manuscript.
Minor concerns: The authors should consistently use one of the alternative names for exenatide, a glucagon-like peptide-1 receptor agonist through the text. In the section 2.3 and Figure 4 they prefer the name exenatide, while in section 2.4 and Figure 5 they use the name 'GLP-1 analog'. Consistent name usage would be preferred.
Response: We used two different peptides (exenatide and a GLP-1 analog). While exenatide has 39 residues (Figure 5a), a GLP-1 analog has a different sequence of a 30-residue peptide with the C-terminal Arg for better cleavage (Figure 6a). We have been using consistent names throughout the manuscript.
Figure 4, panel a contains a typo: 'Exanatide' instead of 'exenatide'.
Response: corrected.
Figure legends of Supplemental Figures should be more detailed, better described.
Response: We added more details in the supplemental Figures.